# Models for estimating and projecting global, regional and national prevalence and disease burden of asthma: protocol for a systematic review

Mohammad Romel Bhuia,[1] Bright I Nwaru,[1,2] Christopher J Weir,[3] Aziz Sheikh[1]

► Prepublication history and additional material are available. To view these files please visit the journal online (http://dx.doi.org/ 10.1136/bmjopen-2016-015441).

[1]Asthma UK Centre for Applied Research, Centre for Medical Informatics, Usher Institute of Population Health Sciences and Informatics, The University of Edinburgh, Edinburgh, UK
[2]School of Health Sciences, University of Tampere, Tampere, Finland
[3]Edinburgh Clinical Trials Unit, Usher Institute of Population Health Sciences and Informatics, The University of Edinburgh, Edinburgh, UK

**Correspondence to**
Mr. Mohammad Romel Bhuia; Mohammad-Romel.Bhuia@ed. ac.uk

## ABSTRACT

**Introduction**  Models that have so far been used to estimate and project the prevalence and disease burden of asthma are in most cases inadequately described and irreproducible. We aim systematically to describe and critique the existing models in relation to their strengths, limitations and reproducibility, and to determine the appropriate models for estimating and projecting the prevalence and disease burden of asthma.

**Methods**  We will search the following electronic databases to identify relevant literature published from 1980 to 2017: Medline, Embase, WHO Library and Information Services and Web of Science Core Collection. We will identify additional studies by searching the reference list of all the retrieved papers and contacting experts. We will include observational studies that used models for estimating and/or projecting prevalence and disease burden of asthma regarding human population of any age and sex. Two independent reviewers will assess the studies for inclusion and extract data from included papers. Data items will include authors' names, publication year, study aims, data source and time period, study population, asthma outcomes, study methodology, model type, model settings, study variables, methods of model derivation, methods of parameter estimation and/ or projection, model fit information, key findings and identified research gaps. A detailed critical narrative synthesis of the models will be undertaken in relation to their strengths, limitations and reproducibility. A quality assessment checklist and scoring framework will be used to determine the appropriate models for estimating and projecting the prevalence and disease burden of asthma.

**Ethics and dissemination**  We will not collect any primary data for this review, and hence there is no need for formal National Health Services Research Ethics Committee approval. We will present our findings at scientific conferences and publish the findings in the peer-reviewed scientific journal.

## BACKGROUND

Asthma is now one of the most common long-term conditions in the world, and it is responsible for substantial morbidity and in some cases mortality.[1] The overall worldwide trend in the prevalence of asthma appears to have plateaued in some parts of the world, while

---

### Strengths and limitations of this study

► To the best of our knowledge, this is the first systematic review to synthesise and critique existing models for estimating and/or projecting the prevalence and disease burden of asthma to scope the relevant evidence base.
► There is no geographical and language limitations.
► Comprehensive and highly sensitive search strategies, identification of studies from leading medical and public health databases, and involvement of a panel of expert will ensure quality of underlying evidence base.
► Panel of experts should be consulted due to lack of standard guidelines for evaluating models.

---

it is still increasing in some countries.[2] Asthma has been ranked as the 14th most important cause of years lived with disability (YLDs) in the world,[3] and it accounts for 1% of all disability-adjusted life years (DALYs) lost globally.[4]

The societal and healthcare costs attributed to asthma are also high across different world regions: for instance, across Europe, the cost of persistent asthma among those aged 15–64 years was estimated in 2010 values at about €19.3 billion[5]; in Asia-Pacific region, the total annual per-patient societal costs of asthma varied from US$184 in Vietnam to US$1189 in Hong Kong (2000 rates).[6] Likewise, at national levels, asthma imposes considerable economic burden to the healthcare system besides its negative impact on the quality of life of individuals and families.[7] [8] For example, recent estimates found that asthma costs at least £1.1 billion per year to UK and its member nations,[9] while in the USA the total cost attributed to asthma in 2007 was estimated at about US$56 billion.[10]

Although varying estimates of asthma prevalence and burden at the national, regional and global levels have been reported in

the published literature,[3 11–16] almost all appear to have major limitations in terms of inadequacy of the analytical approach used and lack of reproducibility.[17–19] There is therefore a need for generating valid and reproducible estimates of disease prevalence and burden of asthma to inform evidence-based policy deliberations. Developing transparent processes for generating the national, regional and global estimates of prevalence and disease burden of asthma will thus enhance reproducibility across settings and will allow reliable projection of future estimates.

In healthcare policy, a model can be defined as a logical mathematical framework or analytical methodology that integrates theories and data to draw inferences regarding parameters of interest to clinicians and decision makers.[20 21] Models are widely used to estimate disease burden,[9 22–24] trend in prevalence[25 26] and future projections[27] of different epidemiological characteristics of asthma. Although current prevalence of asthma can be estimated without applying a model, many studies[15 17 28–31] estimated asthma prevalence applying modelling techniques, particularly with the aim of adjusting for certain population characteristics, such as age, sex, time, geography and other contextual parameters, which may vary across studies. For example, the International Study of Asthma and Allergies in Childhood Steering Committee applied generalised linear mixed model to estimate the global prevalence of asthma in order to adjust for within-country and between-country variations.[15] Adeloye *et al*[17] applied a non-linear model to estimate regional (Africa) prevalence of asthma. The Global Burden of Disease studies also developed some computer based disease modelling tools (such as DisMod, DisMod-MR) to estimate the prevalence and burden of various diseases.[28 32] However, existing models for estimating and projecting prevalence and disease burden of asthma are in most cases poorly described, thereby limiting the opportunity to assess their reproducibility. In order to gain a better appreciation of the performance of existing models and their capacity for reproducibility in estimating the burden of asthma, a systematic appraisal of the underlying evidence base is required.[33]

## Objectives

The aims of this systematic review are to (1) systematically describe and critique the existing models for estimating and/or projecting the global, regional and national prevalence and disease burden of asthma in relation to their strengths, limitations and reproducibility, and (2) determine the appropriate models for estimating and projecting the prevalence and disease burden of asthma.

## METHODS
### Eligibility criteria
#### Types of studies

Any study that developed models for estimating and/or projecting prevalence and disease burden of asthma will be included in this review. Studies that estimated prevalence and disease burden without modelling will be excluded.

Models that estimated individual risk rather than population benefits, such as decision analytic models and individual prognostic models, will be excluded. Moreover, studies with models that simply describe animals, clinical series and cell lines will be excluded. Comparative intervention studies will also be excluded. Potential sources of evidence such as original research articles and review articles, including systematic reviews, meta-analyses and meta-syntheses of observational studies, will be included.

### Participants

Eligible participants in this review will include human populations of any age and either sex.

### Years considered

We will include studies from January 1980 to April 2017. The start date has been set up from the time when modelling techniques started to be applied broadly to study the epidemic of non-communicable diseases.[34]

### Setting

Research articles from any country and any setting (urban/rural) will be included in this review.

### Language

There will be no language restrictions and, where possible, we will translate the literature published in languages other than English.

## Information sources
### Database searches and other sources to identify studies

We will conduct searches to identify both published and unpublished modelling studies in the following electronic databases: Medline, Embase, WHO Library and Information Services (library catalogue of books and reports) and Web of Science Core Collection. The reference lists of all the included papers will be searched for additional studies. We will also contact a panel of experts in an attempt to identify additional unpublished or in progress studies.

## Search strategy

A comprehensive literature search will be undertaken to identify both published and unpublished (grey literature) primary studies as well as reviews. The search strategy has been developed for searching literature in Medline and Embase (see online supplementary appendix) in consultation with a senior medical librarian at The University of Edinburgh, and this will be adapted in searching other databases. The search terms include the concepts of 'modelling', 'prevalence and disease burden' and 'asthma'.

## Study records
### Data management

The retrieved records from all databases will be exported to EndNote Library, which will be used throughout the review for study screening, deduplication and overall management of the retrieved records.

### Selection process

Two reviewers will independently check and screen the titles and abstracts of identified articles against the inclusion criteria. Full-text copies of potentially relevant studies will be obtained and assessed by two independent reviewers on the basis of their eligibility for inclusion. Any discrepancies will be resolved by discussion, and disagreements will be arbitrated by a third reviewer.

### Data extraction

A data extraction form will be used to extract relevant data from included studies. We have developed a draft data extraction form. During the review process, this draft will be refined and the data extraction form will be updated accordingly. The data extraction form will be pre-piloted prior to full use in the review. Data extraction will be performed independently by two reviewers.

### Data items

Information regarding different components of the models will be recorded to get a comprehensive picture of the models. The following data items will be extracted from each study: authors' names; publication year; study aims; data source and time period; study population; asthma outcomes (prevalence/disease burden); study methodology; model type; model settings; model formulation (structure, specification, assumptions, methods of model derivation, methods of parameter estimation and/or projection, theoretical basis of the models) study variables; availability of data and codes; findings from the models; model fit information; key findings of the study; and identified research gaps. Information regarding the model availability, transparency, sensitivity analysis, model validation, addressing missing data, policymakers' involvement, dissemination and expert involvement, limitation discussed and reproducibility of the model will also be extracted. Descriptive tables will be used to tabulate these items. The systematic review will be reported following the guidelines of the Preferred Reporting Items for Systematic review and Meta Analysis (PRISMA) checklist (see online supplementary material).[35]

### Outcomes and prioritisation

The outcomes that are of interest include prevalence and disease burden of asthma. There are various measures available to quantify disease burden. All the established measures of disease burden will be considered in this review. Primary and secondary outcomes are categorised as follows.

#### Primary outcomes

1. Prevalence of asthma.
2. Different measures of disease burden of asthma. The measures are: DALYs, YLDs, mortality, healthcare cost (cost of illness, drug cost, hospital cost/hospitalisation cost), life expectancy, primary care, ambulatory care, emergency visit, absentees, years life lost, potential years of life lost, healthy years of life lost, active life expectancy, disability-free life expectancy, disability-adjusted life expectancy, healthy life expectancy, and so on.

#### Secondary outcomes

1. Incidence of asthma.

### Risk of bias in individual studies

To the best of our knowledge, there is no existing quality appraisal tool to assess quality of models. So we have drawn on first principles and adapted relevant sections from pertinent reporting guidelines[36] and other guidelines for good practice in modelling studies[20 21 37 38] to develop our own model evaluation framework. This will involve independent assessment of the strengths and limitations of the models on the basis of model structure, specification, assumptions, sensitivity analysis, model validation, dealing with missing data, theoretical basis of the models, incorporation of confounding factors and lag times, and whether potential methodological limitations are described. Reproducibility of the model will be assessed on the basis of availability of the models, data, codes and methods of parameter estimation.

To evaluate the models used in included studies and to identify the best models, we have prepared a checklist of items and formulated a scoring strategy (see online supplementary appendix) that we will use for these purposes. Prior to use of the checklist, we plan to consult with a panel of experts in the field of modelling studies to gain their insights and criticisms of the checklist; we will then integrate feedback collated in preparing the final version of the checklist to be used in our study.

### Data synthesis

A tabular summary of the data will be presented to summarise overall evidence. A detailed critical narrative synthesis of the models will be undertaken regarding their strengths, limitations and reproducibility.

## PROTOCOL REGISTRATION

A detailed protocol for the systematic review will be registered with the International Prospective Register of Systematic Reviews prior to commencing the review according to the PRISMA-Protocols 2015 statement.[39]

## CONCLUSIONS

To the best of our knowledge no review has been undertaken yet to appraise the models for estimating and projecting the global, regional and national prevalence and disease burden of asthma. This systematic review is therefore the first study to synthesise existing models for estimating and projecting prevalence and disease burden of asthma. The review will also map the appropriate models that will subsequently be used to obtain current estimates and project future trend of global, regional and national prevalence and disease burden of asthma.

## ETHICS AND DISSEMINATION

We will not collect any primary data for this review, and hence there is no need for formal National Health Service Research Ethics Committee review. This work is however subject to Institutional Review Board oversight by The University of Edinburgh's Centre for Population Health Sciences. Findings from the review will be presented at scientific conferences and be published in the peer-reviewed scientific journal.

**Acknowledgements** We are thankful to Marshall Dozier, Senior Liaison Librarian for the College of Medicine and Veterinary Medicine, The University of Edinburgh, for her support in developing the search strategies. We are also thankful to Dr Susannah McLean for providing necessary suggestions and resources regarding modelling studies, Dr Niall Anderson for his suggestions in developing scoring framework to appraise the models, Andrew Stoddart for suggesting search terms related to economic burden of asthma, and the Asthma UK Centre for Applied Research for providing platform to develop this protocol.

**Contributors** AS conceived the idea for this work. MRB drafted the protocol under the supervision of AS, BIN and CJW. The draft was revised according to several rounds of critical comments from AS, BIN and CJW. All the authors will be involved in the systematic review process.

**Funding** MRB is supported by the College of Medicine & Veterinary Medicine, The University of Edinburgh, and the Farr Institute, UK. BIN and AS are also supported by the Farr Institute, UK. The Farr Institute is funded by a consortium of funders led by the Medical Research Council (MRC).

**Competing interests** None declared.

**Provenance and peer review** Not commissioned; externally peer reviewed.

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
