## [Reviewer comments · BMJ Open]

ARTICLE DETAILS

TITLE (PROVISIONAL)	Models for estimating and projecting global, regional and national prevalence and disease burden of asthma: protocol for a systematic review
AUTHORS	Bhuia, Mohammad Romel; Nwaru, Bright; Weir, Christopher; Sheikh, Aziz

VERSION 1 - REVIEW

REVIEWER	Onno van Schayck Maastricht University The Netherlands
REVIEW RETURNED	22-Dec-2016

GENERAL COMMENTS	The paper 'Models for estimating and projecting global, regional and national prevalence and disease burden of asthma: a scoping review protocol' is a protocol paper on a scoping review which aims to describe and select well-developed models for estimating and projecting the prevalence and disease burden of asthma. The scoping review seeks to find out whether existing models are appropriate and can be reproduced. The main strength of the papers is its relevance as the authors aim to determine whether the estimates being made on the burden of asthma are accurate. Furthermore, the results of this study may facilitate standardization in estimating disease prevalence and disease burden. Nevertheless, I have a few comments: 1. Abstract The abstract lacked information about data extraction. It was not clear which information is going to be retrieved. What are the extraction methods? I could therefore not grasp from the abstract how the authors will make their value judgments. 2. Introduction The authors aim to examine whether existing models are appropriate for estimating the current prevalence of asthma (1) and/or for projecting the future prevalence and associated disease burden (2). I think these are two different questions with different approaches. The appraisal on estimating asthma prevalence will mainly involve the selection of data sources (self-report, GP registry etc.). Because models are often used to estimate disease burden rather than assessing current prevalence rates, I doubt whether the scoping review will provide an answer to the first question. The introduction focuses mainly on the second problem because the selected model type, methods and the way these are reported may highly determine whether results are valid and reproducible. If this assumption is not correct it should be better clarified in the
--

	Introduction. 3. Methods A scoping review is very useful to synthesize studies that have different study designs. However, I am not convinced that a scoping review provides the best means to assess which models are most appropriate because a scoping review is generally not aimed at assessing quality (JBI guidance , Tricco 2016). The authors state that quality checklists and risk of bias tools will not be used. To support their value judgments I think that the authors should consult quality checklists. With regard to data extraction, the authors mention that key findings are tabulated. I expect this will only involve study results, but the authors refer to general study characteristics, methodological aspects and results. This needs some clarification. Last, it is mentioned that papers from any setting will be included. It is unclear what is meant by setting (e.g. geographical location, healthcare settings).
--	---

REVIEWER	George V. Guibas Division of Infection, Immunity and Respiratory Medicine University of Manchester UK
REVIEW RETURNED	31-Dec-2016

GENERAL COMMENTS	The authors state their intention to describe and critique the existing models that are used to estimate and project the prevalence/burden of asthma, and to determine which are more appropriate. This is an interesting and worthwhile pursuit. The methods for the literature search appear to be rigorous, and the search terms in the supplementary material are appropriate. The authors do well in describing how they will search for appropriate studies. Some additional information on how they will evaluate the models would, however, be welcome. I have no major concerns, as this manuscript is generally well-written. Where the authors state that 'a primary limited search will be carried out in two databases namely MEDLINE and EMBASE' It would be nice if there is a clarification of whether the authors intend to use the same search terms that are included in the supplementary material.
--

VERSION 1 – AUTHOR RESPONSE

REVIEWER: 1

Reviewer Name: Onno van Schayck

Institution and Country: Maastricht University, The Netherlands Please state any competing interests or state 'None declared': None

Please leave your comments for the authors below

The paper 'Models for estimating and projecting global, regional and national prevalence and disease burden of asthma: a scoping review protocol' is a protocol paper on a scoping review which aims to describe and select well-developed models for estimating and projecting the prevalence and disease burden of asthma. The scoping review seeks to find out whether existing models are appropriate and

can be reproduced.

The main strength of the paper is its relevance as the authors aim to determine whether the estimates being made on the burden of asthma are accurate. Furthermore, the results of this study may facilitate standardization in estimating disease prevalence and disease burden. Nevertheless, I have a few comments:

1. Abstract

Comment:

The abstract lacked information about data extraction. It was not clear which information is going to be retrieved. What are the extraction methods? I could therefore not grasp from the abstract how the authors will make their value judgments.

Response:

We have now added details in the abstract about data items (which information is going to be retrieved), data extraction method, and quality appraisal framework for evaluating models (page 2, lines 13-22).

2. Introduction

Comment:

The authors aim to examine whether existing models are appropriate for estimating the current prevalence of asthma (1) and/or for projecting the future prevalence and associated disease burden (2). I think these are two different questions with different approaches. The appraisal on estimating asthma prevalence will mainly involve the selection of data sources (self-report, GP registry etc.). Because models are often used to estimate disease burden rather than assessing current prevalence rates, I doubt whether the scoping review will provide an answer to the first question. The introduction focuses mainly on the second problem because the selected model type, methods and the way these are reported may highly determine whether results are valid and reproducible. If this assumption is not correct it should be better clarified in the Introduction.

Response:

Although current prevalence can be estimated without applying a model, many studies estimated asthma prevalence applying modelling techniques, particularly with the aim of adjusting for certain vital population characteristics, such as age, sex, time, geography, and other contextual parameters that may vary across studies. For example, the International Study of Asthma and Allergies in Childhood (ISAAC) Steering Committee (The European Respiratory Journal 1998;12(2):315-35) applied generalised linear mixed model to estimate the global prevalence of asthma in order to adjust for within-country and between-country variations. Adeloje et al (Croatian Medical Journal 2013;54(6):519-31) applied a non-linear model to estimate prevalence of asthma for Africa region. The Global Burden of Disease (GBD) studies also developed some models (DisMod, DisMod II, DisMod-MR, DisMod-MR 2.1) for estimating disease prevalence. In the light of the reviewer's helpful comment, we have expanded the Introduction to provide justification for reviewing models for estimating asthma prevalence (page 5, lines 6-18).

3. Methods

Comment:

A scoping review is very useful to synthesize studies that have different study designs. However, I am not convinced that a scoping review provides the best means to assess which models are most appropriate because a scoping review is generally not aimed at assessing quality (JBI guidance , Tricco 2016). The authors state that quality checklists and risk of bias tools will not be used. To support their value judgments I think that the authors should consult quality checklists.

Response:

We thank the reviewer for raising this important issue and agree with the reviewer that a scoping review will not sufficiently address our study questions, particularly with regards to assessing quality of the underlying evidence. Consequently, we have now changed the manuscript from a scoping review to a full systematic review as this will then enable us to undertake full quality appraisal of the studies and be able to assess the strengths, limitations and reproducibility of the models in included studies. Although there is no existing quality assessment tool for appraising modelling studies, we have now designed a preliminary quality assessment checklist and scoring framework that we will employ in appraising the studies. We plan to further refine this checklist after consultation with experts in the field. These revisions are marked in the manuscript (page 10, lines 13-24 and page 11, lines 1-6).

Comment:

With regard to data extraction, the authors mention that key findings are tabulated. I expect this will only involve study results, but the authors refer to general study characteristics, methodological aspects and results. This needs some clarification.

Response:

We have now removed the term 'key findings' and mentioned all the data items that will be extracted from each of the studies (page 9, lines 5-16).

Comment:

Last, it is mentioned that papers from any setting will be included. It is unclear what is meant by setting (e.g. geographical location, healthcare settings).

Response:

We have corrected this by specifying the term 'any setting (urban/rural)' (page 7, line 7).

REVIEWER: 2

Reviewer Name: George V. Guibas

Institution and Country: Division of Infection, Immunity and Respiratory Medicine, University of Manchester, UK

Please state any competing interests or state 'None declared': None Declared

Please leave your comments for the authors below

The authors state their intention to describe and critique the existing models that are used to estimate and project the prevalence/burden of asthma, and to determine which are more appropriate. This is an interesting and worthwhile pursuit.

Comment:

The methods for the literature search appear to be rigorous, and the search terms in the supplementary material are appropriate. The authors do well in describing how they will search for appropriate studies. Some additional information on how they will evaluate the models would, however, be welcome.

Response:

We thank the reviewer for the positive feedback. Additional information on how the models will be evaluated has now been added to the manuscript (page 10, lines 13-24 and page 11, lines 1-6). We have designed a quality assessment checklist and scoring framework in order to evaluate the models and determine the appropriate models for estimating and projecting the prevalence and burden of asthma.

Comment:

I have no major concerns, as this manuscript is generally well-written. Where the authors state that 'a primary limited search will be carried out in two databases namely MEDLINE and EMBASE' It would be nice if there is a clarification of whether the authors intend to use the same search terms that are included in the supplementary matterial.

Response:

As noted above, we now plan to undertake quality assessment of the models identified. We have therefore now changed the manuscript from a scoping review protocol to a systematic review protocol. We will as a result now carry out a comprehensive literature search instead of the three-step search that was proposed for scoping review. We have clarified this in the manuscript that we intend to use the same search terms that are included in the supplementary Appendix (page 8, lines 3-4).

VERSION 2 – REVIEW

REVIEWER	Onno van Schayck University Maastricht The Netherlands
REVIEW RETURNED	07-Feb-2017

GENERAL COMMENTS	The authors have satisfactorily dealt with all issued raised.
---